# Bamboo Scaffolding as a European Promising Opportunity: A Structural Feasibility Study

**Davide Altieri and Luisa Molari \*** 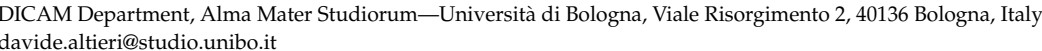

DICAM Department, Alma Mater Studiorum—Università di Bologna, Viale Risorgimento 2, 40136 Bologna, Italy; davide.altieri@studio.unibo.it
\* Correspondence: luisa.molari@unibo.it

**Abstract:** There is an increasing need for the use of materials with low carbon emissions, especially in the construction sector, which is responsible for a large amount of emissions. In this perspective, bamboo can play a crucial role; it grows very fast and is an effective carbon sink, having good mechanical properties. It has been used for millennia in specific contexts and territories, such as scaffolding in Asia. This paper aims to demonstrate how bamboo scaffolding can be a viable option in Europe as well. Two prototypes of scaffolding are calculated for the refurbishment of two- or three-story buildings, a common typology in European urban and extra-urban residential areas. The dimensions of the considered scaffolding are 1.8 m and 1.0 m for bay lengths and 1.5 m and 0.8 m for bay lifts. The bamboo considered in the analysis grows in temperate climates, dimensionally smaller in diameter and thickness than tropical ones. Connections play a crucial role, and this paper proposes simple steel connections that are easily removable and suitable for standardized assembly processes already available on the market.

**Keywords:** bamboo; scaffolding; European bamboo; structural analysis; connections

## 1. Introduction

The search for sustainable materials is a priority due to the non-postponable concern about the impact of human activities on the environment, climate change, and social aspects [1]. The high environmental impact caused by the construction sector is widely recognized [2], and bamboo can be the material of its green transition.

Bamboo has over 1600 species that can grow at different longitudes and altitudes in the world. As stated in the Global Forest Resources Assessment 2020 of the Food and Agriculture Organization of the UN (FAO) [3], bamboo covers about 35 million hectares of land across Africa, Asia, and the Americas, with an increase of 50 percent in bamboo area between 1990 and 2020.

Bamboo is one of the fastest-growing plants [4], and it is an effective carbon sink, accumulating carbon quicker than other plants. Yuen et al. [5] showed that a Moso bamboo forest in China can sequester 5.1 t/ha of carbon per year, 33% more than a tropical mountain rainforest. Xu et al. [6] reported sequestration of 6.0–7.6 t/ha of carbon each year in the harvested Moso bamboo, 3.7 times higher than that of *Pinus taeda*.

Once mature, bamboo poles can be selectively harvested and used as a structural durable product, which locks carbon for the duration of the structure's life [7]. Furthermore, the use of bamboo as a structural material can also avoid the carbon produced by more conventional materials. In Europe, scaffolding is typically made of steel. Laleicke et al. in [8] show the comparison in terms of Life Cycle Assessment (LCA) [9] cradle-to-gate between scaffolding made of steel and bamboo. The 'gate' is considered the state after ten construction/deconstruction cycles of the scaffolding system. Considering some assumptions about the provenience of the raw materials and of the production processes, the paper shows a significant difference in terms of the mass of carbon dioxide ($CO_2$)

equivalent at the cradle (5.41 kg of $CO_2$ per functional unit for steel with respect to $-223$ kg of $CO_2$ per functional unit for bamboo) and the gate (264.7 kg of $CO_2$ per functional unit for steel with respect to 2.0 kg of $CO_2$ per functional unit for bamboo). The values show the different impacts of the production processes. However, it is important also to take into consideration other indicators like resource depletion and waste generation [10]. In this perspective, a crucial rule is played by the durability of bamboo.

A possible scaffolding unit consists of two layers composed of standards (the vertical elements), ledgers (the horizontal element), and face braces (the diagonal elements in the layer plane). The two layers are connected by transversal elements named transoms (the horizontal elements) and the transverse braces (the diagonal elements). The distance between the standards in the layers is called the bay length, while the distance between the standards in the transversal direction is named the bay width. The height between the platforms is called the lift height. All the nomenclature used is represented in Figure 1. There are also several elements of protection such as the guard rails or the toe boards.

Bamboo scaffolding has been traditionally used in many Asian countries for centuries. Using bamboo in scaffolding has advantages such as lower environmental impact compared to steel scaffolding [11,12] and cost-effectiveness, which is particularly important for construction projects in developing countries.

Bamboo scaffolding has been used in the Hong Kong region for centuries. There are typically three types of bamboo scaffolding [13–15]: two-level scaffolding, suspended scaffolding, and scaffolding for advertising. Each of these follows its design criteria regarding the species of bamboo and the geometry of the entire structure. Among these, the type of scaffolding most similar to that commonly used in Europe is the double-layer scaffolding. This structure uses two types of bamboo species: *Bambusa pervariabilis* with a nominal external diameter not less than 40 mm, called Kao Jue, and *Phyllostachys edulis* with a diameter not less than 75 mm, called Mao Jue. In both cases, the culms must be between 3 and 5 years old and must have been properly dried indoors for at least 3 months before use. The internal layer of the scaffolding is placed 20–25 cm from the facade of the building, and the external layer is placed at a distance of 60 cm from the internal layer. As for the outermost layer, the standards are made with Mao Jue elements and are placed at a maximum distance of approximately 1.3 m (bay length). The ledgers are Kao Jue elements with small diameters, except the one placed at the base, for which a larger diameter is chosen. A bracing system is added by positioning two Kao Jue elements in an "X" shape with an inclination between 45 and 60°. Each bracing element must be appropriately linked to all the vertical and horizontal elements that constitute the layer. Furthermore, each outermost main standard is connected to the facade of the building through steel tie rods. Regarding two overlapping parallel elements, the guidelines specify that the overlap length can be between 1.5 and 2 m for the ledgers and that the maximum distance between two overlap positions is not greater than 3 m [14]. Connections are a crucial element of bamboo scaffolding. In Hong Kong, the knots of the structure's elements are made using nylon strips tied appropriately using specific techniques. Three types of knots are distinguished: basic knot, retained knot, and reinforced knot. The basic knot, mainly used to connect a standard and a ledger or bracing element, presents five turns of the strip around the rods (see Figure 2). In the retained and reinforced knots, which are used especially for bracing elements that have an inclination, the knots are also closed orthogonally at the first turn, with an additional nylon strip or galvanized wire. In the reinforced knot, there are more turns, and it is usually used for the most stressed elements [14].

The techniques used in Hong Kong with nylon strips cannot be adopted in Europe because of the lack of technical expertise and the inability to achieve a measurable and standardizable level of safety [16,17]. Despite the widespread use of bamboo scaffolding in Asia, the diffusion of bamboo construction in Europe has been limited for several reasons: the lack of regulations and safety standards, safety concerns, and the requirement of skilled artisans for the assembly and disassembly process. The lack of regulations and safety standards is slowly being addressed, and the last publication of ISO ISO22156 marked a

milestone in the bamboo design guideline [18,19]. Safety concerns and the need for skilled artisans can be overcome with the design. Furthermore, periods of high demand, such as the current situation in Italy with a government incentive for building retrofitting, should be exploited to expand the knowledge and the use of alternative materials like bamboo.

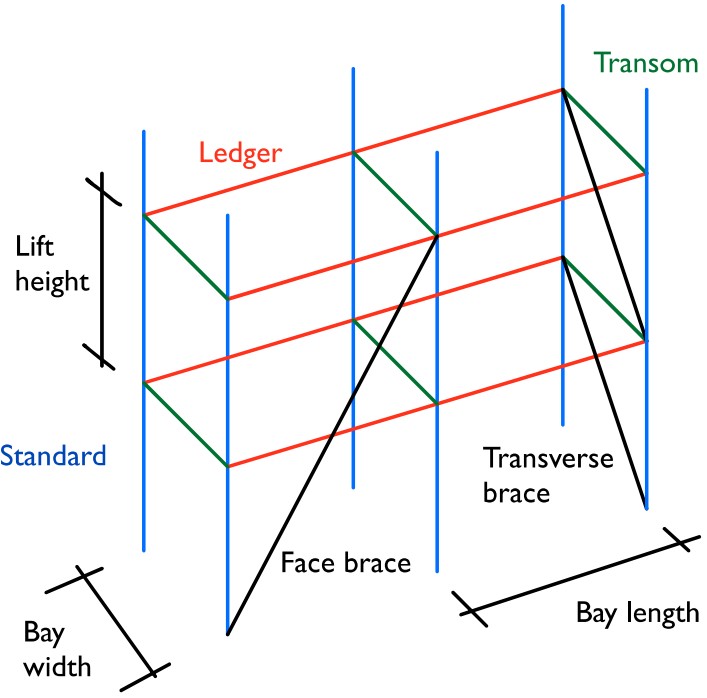

**Figure 1.** Nomenclature of the scaffolding system.

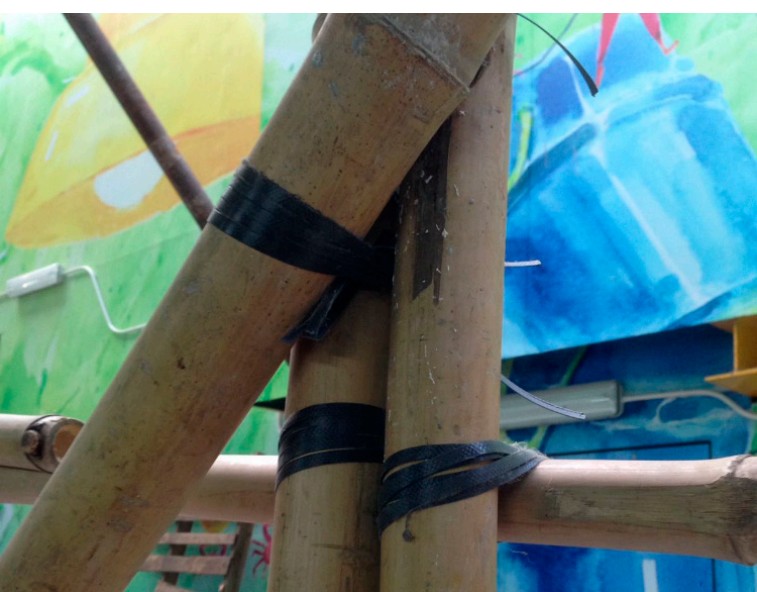

**Figure 2.** The connection with 5 turns of the strip around the rods.

This study aims to adapt the Asian experience to the Italian and European context. Following European and Italian regulations, two scaffolding prototypes are designed. Engineered steel connections are proposed, after some preliminary laboratory resistance tests. The bamboo used in the paper is bamboo that can grow, for size and species, in a temperate climate such as the Italian or European one [20].

## 2. Design of the Scaffolding

The design and construction of scaffolding are influenced by the environmental context (i.e., the site and the location external or internal to the building) and the type of work to be carried out, which can vary from simple ordinary maintenance interventions to the construction of new structures. This study focuses on two modules considering two and three bay lengths and is limited to scaffolding with a double-layer structure with relatively modest actions. The scaffolding is built near the facade of a building and is intended only for maintenance and/or renovation works. It can be used on two or three-story buildings, typical of a historic urban center or extra-urban residential area. The guidelines EN12810-1:2004 [21] and EN12810-2:2004 [22], together with ISO22156:2021 [18] and the Italian construction regulations NTC2018 [23], are followed to establish both the geometric limits and the load conditions and combinations for the structural analysis. The bracing scheme reported in Figure 3 is adopted, where diagonals have been inserted into all modules on the external layer of the scaffolding. An exception is a portal at the base in which no diagonal elements have been inserted to make the interior of the structure accessible. On the innermost facade, the diagonals were placed only on a portion of the structure. The structure was braced similarly in the transversal direction to the facade of the building: diagonals were considered only on the portals located at the two ends of the scaffolding.

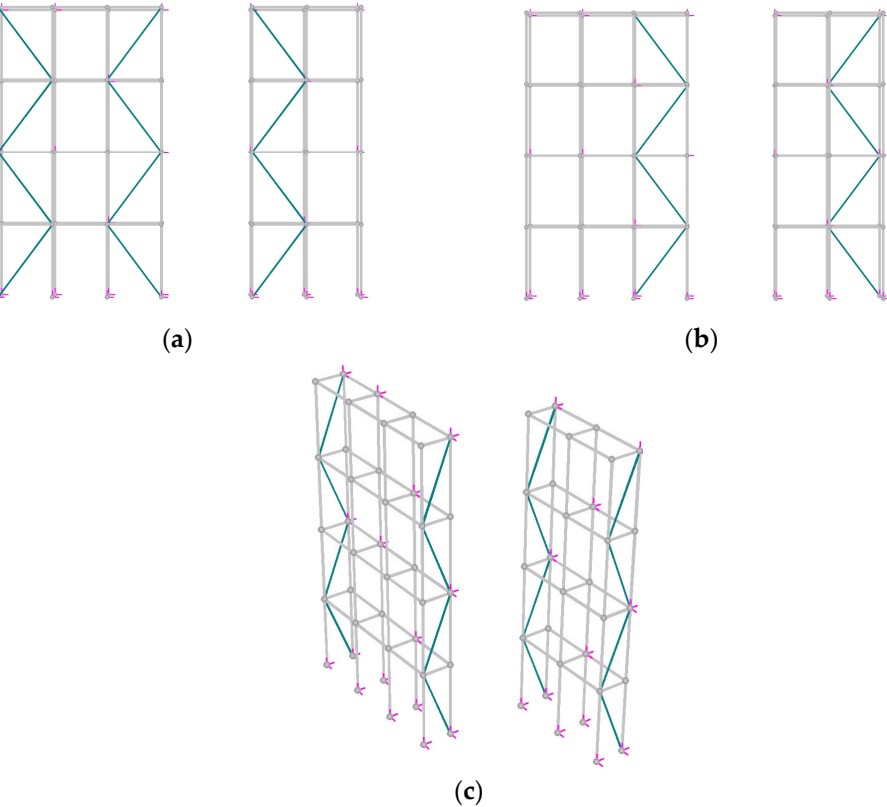

(a)          (b)

(c)

**Figure 3.** A bracing system of the two- and three-bay-length modules. (**a**) External layer, (**b**) internal layer, and (**c**) lateral system.

*Connections*

Connections are a crucial point for frame structures made of bamboo culms. The basic idea is to use sufficiently resistant and rigid joints that can be assembled and disassembled in a short time and that do not require specialized workers. To simplify feasibility and improve the performance and effectiveness of the joints, steel joints are introduced. Different joints are designed for the different mutual orientations of the culms. All types of joints have in common a steel connection and the use of the steel throat band cable tie.

The orthogonal connection is the most widespread connection within the structure. We chose to use the BAMBOTIX-X joint produced by German company CONBAM (represented in Figure 4), which is suggested by the ISO22156:2021 standard in Annex D, Examples and classification of bamboo connections and joints, Figure D.5. Given the absence of mechanical characteristics of the joints, some preliminary mechanical tests were carried out.

The tests reported are preliminary and aim to show the feasibility of the joint. Figure 4 shows the setup to test two connections simultaneously to limit the eccentricity. We tested two BAMBOTIX-X connections tightened by two steel throat band cable ties. In the tests, the diameter of the vertical culm is approximately 60 mm, and the two horizontal culms have a diameter of 50 mm. The test consists of compressing the vertical culm while the horizontal culms are placed on two supports with a span of 15 cm. The tests were conducted in the LISG laboratory of the University of Bologna on a Galdabini PMA10 press machine. Various tests were carried out by modifying the tightening torque (evaluated using a torque wrench) and the thickness of the steel throat band cable tie. The first test was carried out using a 9 mm steel throat band cable tie available on the market, tightened by a 5 Nm tightening moment. The other tests were carried out using a steel throat band cable tie with a thickness of 12.7 mm with the following three different tightening moments: 5 Nm, 7 Nm, and 12 Nm. For the tests with a thicker steel throat band cable tie, it was necessary to widen the slot of the connector, which measured 10 mm, as shown in Figure 5.

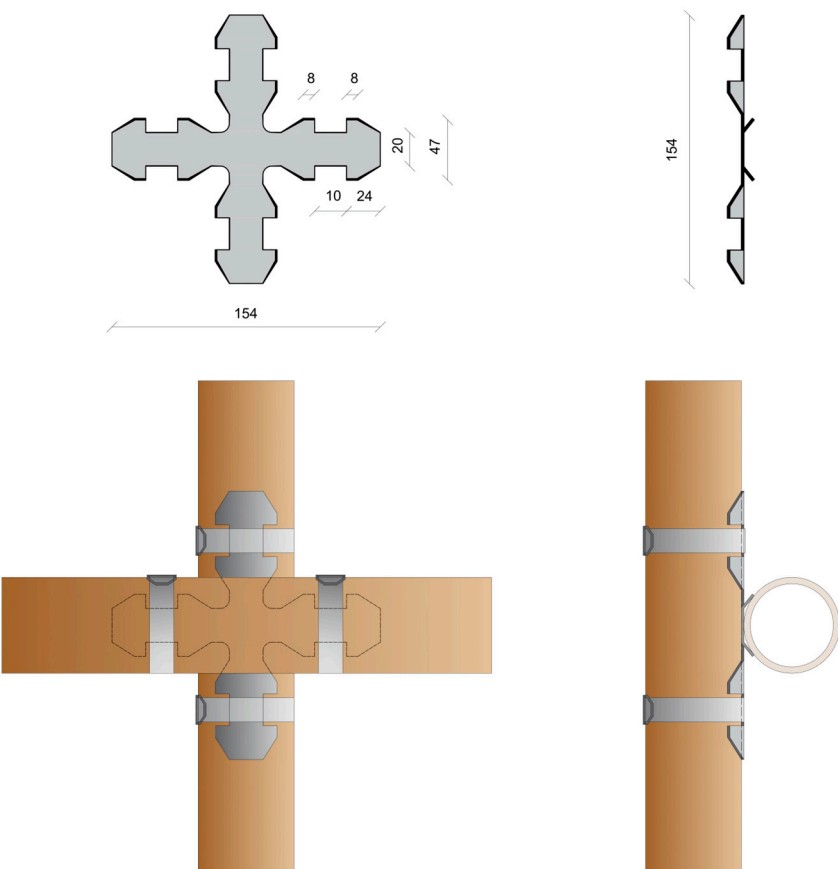

**Figure 4.** Connection with orthogonal culms using the BAMBOTIX-X by CONBAM.

Figure 6 shows the load–displacement diagrams of the tests. Inspecting the graphs reveals, as expected, the increasing resistance of the connection by increasing the size of the clamp and the tightening torque value. With a tightening torque of 12 Nm and the clamp at 12.7 mm, a load of 1.95 kN is achieved as reported in Table 1. An important aspect that is evident from the curves in Figure 6 is the ductility of the joint.

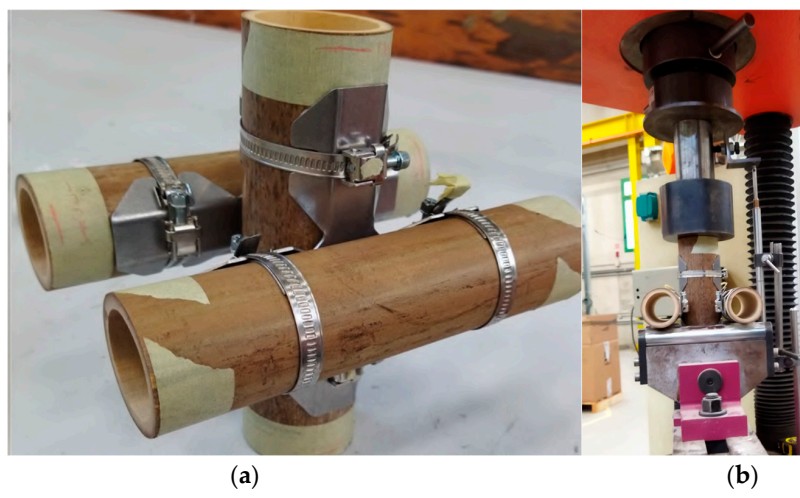

<center>(<b>a</b>)            (<b>b</b>)</center>

**Figure 5.** (**a**) Tested system with two connections. (**b**) Setup of the test.

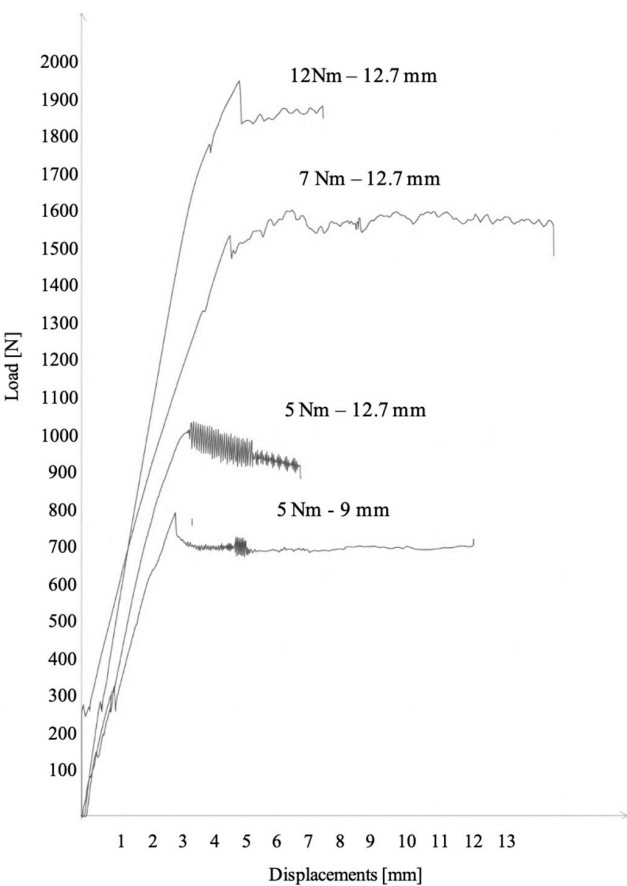

**Figure 6.** Load–displacement curves.

**Table 1.** Maximum load reached on the different tests.

| Tightening Moment [Nm] | Thickness of Throat Band Cable Tie [mm] | Load [kN] |
| --- | --- | --- |
| 5 | 9.0 | 0.77 |
| 5 | 12.7 | 0.99 |
| 7 | 12.7 | 1.55 |
| 12 | 12.7 | 1.95 |

The modified BAMBUTICK connector represented in Figure 7 is used for the joints between inclined culms. Instead of a crossed-shaped single piece of steel, two pieces are designed that can rotate to each other around a pin.

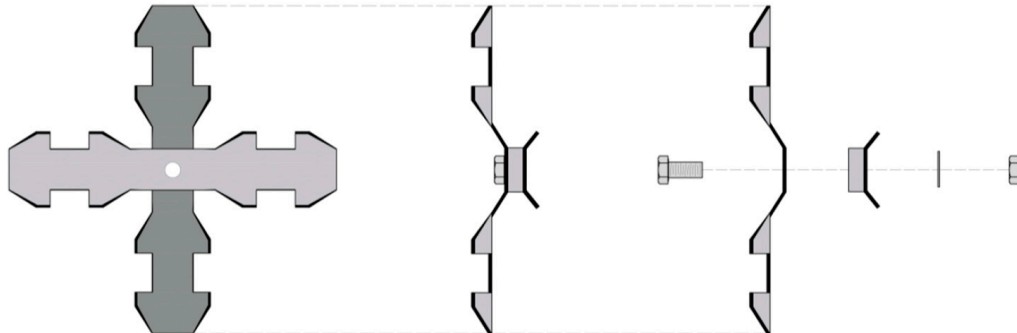

**Figure 7.** Sketch of the connections for inclined culms.

Figure 8 shows the connection between two parallel culms such as the two consecutive elements of a *standard*. The system is composed of two steel pipes welded with a support plate. The pipes are placed inside the culms, and the plate acts as support. An external connector consisting of a single vertical piece of steel is added and clamped to the culm with two *throat band cable ties*, to avoid any movement between the steel pipe and the bamboo culm, which has a variable cross-section. The supporting plate has two holes to let the connectors pass.

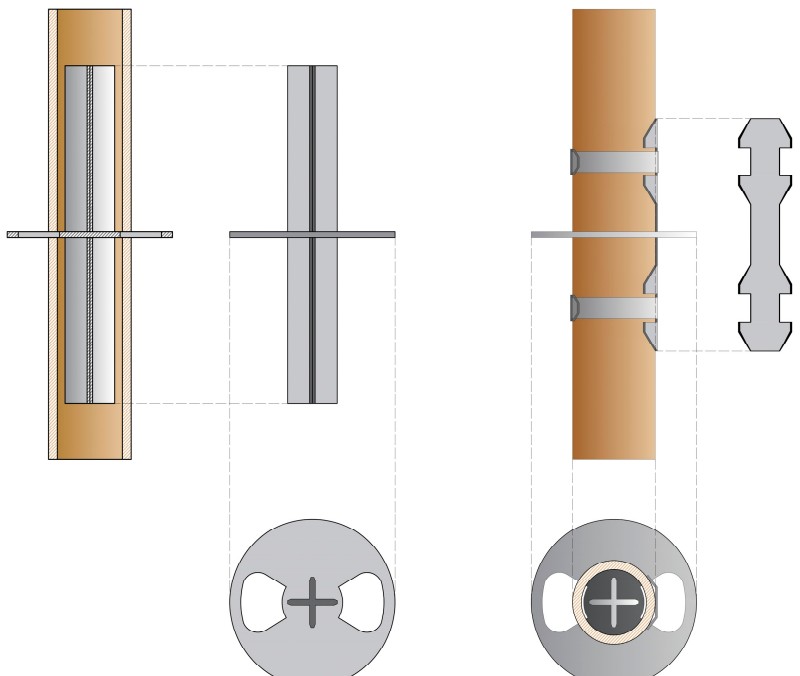

**Figure 8.** Connection between culms with the same orientation.

For the nodes at the base, a circular plate of 15 cm in diameter is welded to a steel tube, which is directly inserted inside the culm to distribute the action to the ground. The system is represented in Figure 9.

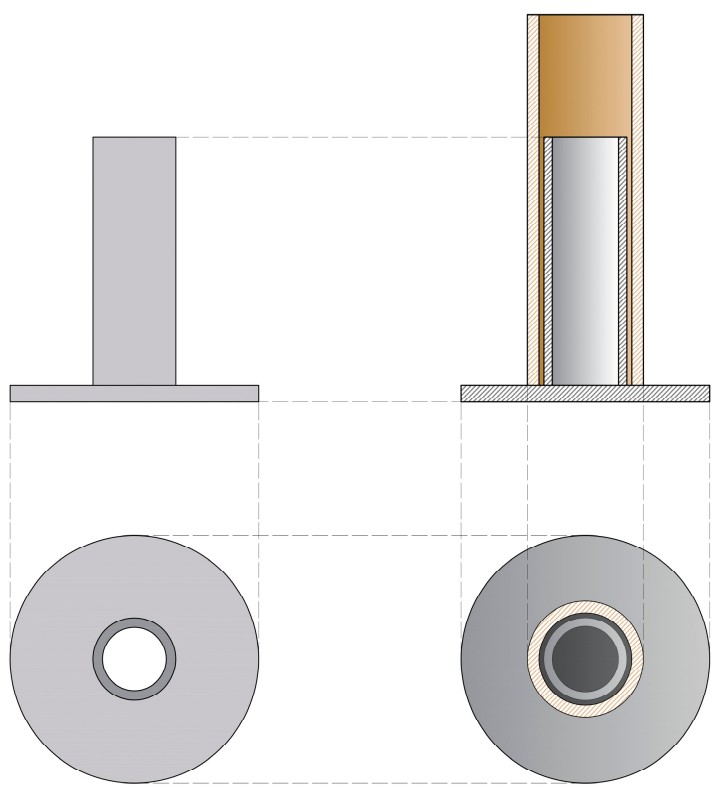

**Figure 9.** Basal part of the scaffolding.

### 3. Case Studies: Prototype 1 and 2 Dimensioning and Verification

Two prototypes with different dimensions were considered, the first with a bay length of 1.80 m and the second of reduced dimensions equal to 1.50 m (in both cases, the bay lengths are greater than 1.40 m, which is the minimum distance suggested by EN12810-1:2004 for the pedestrian crossing).

Furthermore, in the first prototype, the bay width is 1 m, while in the second prototype, the bay width is 0.8 m. The maximum height of the facade has been limited to 10 m with four work lift heights at a 2 m distance. Considering the presence of the elements used on the fourth floor for the safety system, the total height of the scaffolding is 9.4 m. Table 2 summarizes the geometric characteristics adopted for the two scaffoldings.

**Table 2.** Geometric properties of Prototypes 1 and 2.

| Structural Elements | Prototype 1 | Prototype 2 |
|---|---|---|
| Bay length [m] | 1.8 | 1.5 |
| Bay lift [m] | 1.0 | 0.8 |
| Height lift $h$ [m] | 2.0 | 2.0 |
| Total height $H$ [m] | 9.4 | 9.4 |

For both prototypes, two models were analyzed, one with two spans and one with three spans, to simulate the minimum module that needs to be repeated.

### 4. Materials

The bamboo used for the project is Phyllostachys Edulis with the characteristics reported in Table 3 [20], considering the characteristic values calculated by [24] reported in Table 4.

**Table 3.** Characteristic values: $f_{t,0,k}$, tension strength parallel to the fibers; $f_{c,0,k}$, compression strength parallel to the fibers; $f_{c,90,k}$, compression strength orthogonal to the fibers; $f_{m,0,k}$, flexural strength parallel to the fiber; $f_{m,90,k}$, flexural strength orthogonal to the fibers; $f_{v,k}$, shear strength; $E_{t,0,k}$, Young modulus in tension parallel to the fibers; $E_{c,0,k}$, Young modulus in compression parallel to the fibers; $E_{c,90,k}$, Young modulus in compression orthogonal to the fibers; $E_{m,0,k}$, flexural Young modulus parallel to the fibers; $G_k$, tangential modulus.

| Resistance | | Young Moduli | |
|---|---|---|---|
| $f_{t,0,k}$ [Mpa] | 98.4 | $E_{t,0,k}$ [Gpa] | 14.61 |
| $f_{c,0,k}$ [Mpa] | 49.7 | $E_{c,0,k}$ [GPa] | 13.26 |
| $f_{c,90,k}$ [Mpa] | 10.2 | $E_{c,90,k}$ [Gpa] | 13.00 |
| $f_{m,0,k}$ [Mpa] | 65.8 | $E_{m,0,k}$ [Gpa] | 3.04 |
| $f_{m,90,k}$ [MPa] | 17.2 | $G_k$ [Gpa] | 2.52 |
| $f_{v,k}$ [MPa] | 7.9 | | |

**Table 4.** Design values for the different loads in the second service class.

| | $C_{DF}$ | $FS_M$ | Characteristic Values | | $C_{DE}$ | Characteristic Values |
|---|---|---|---|---|---|---|
| $f_{t,0,d}$ [Mpa] | 0.55 | 2 | 24.3 | | 0.45 | 6.57 |
| | 0.65 | 2 | 28.8 | $E_{t,0,d}$ [Gpa] | 0.95 | 13.88 |
| | 0.85 | 2 | 37.6 | | 1.00 | 14.61 |
| $f_{c,0,d}$ [Mpa] | 0.55 | 2 | 12.3 | | 0.45 | 5.97 |
| | 0.65 | 2 | 14.5 | $E_{c,0,d}$ [Gpa] | 0.95 | 12.60 |
| | 0.85 | 2 | 19.0 | | 1.00 | 13.26 |
| $f_{c,90,d}$ [Mpa] | 0.55 | 4 | 1.3 | | 0.45 | 5.85 |
| | 0.65 | 4 | 1.5 | $E_{m,0,d}$ [Gpa] | 0.95 | 12.35 |
| | 0.85 | 4 | 1.9 | | 1.00 | 13.00 |
| $f_{m,0,d}$ [MPa] | 0.55 | 2 | 16.3 | | 0.45 | 1.37 |
| | 0.65 | 2 | 19.2 | $E_{m,90,d}$ [Gpa] | 0.95 | 2.89 |
| | 0.85 | 2 | 25.2 | | 1.00 | 3.04 |
| $f_{m,90,d}$ [MPa] | 0.55 | 2 | 4.3 | | 0.45 | 1.13 |
| | 0.65 | 2 | 5.0 | $G_d$ [Gpa] | 0.95 | 2.39 |
| | 0.85 | 2 | 6.6 | | 1.00 | 2.52 |
| $f_{v,d}$ [MPa] | 0.55 | 4 | 1.0 | | | |
| | 0.65 | 4 | 1.2 | | | |
| | 0.85 | 4 | 1.5 | | | |

According to ISO22156:2021 [18], each design resistance must be evaluated by multiplying the respective characteristic value at the 5% fractile and with a confidence of 75%, $f_{i,k}$, by some correction coefficients:

$$f_{i,d} = \frac{f_{i,k} \cdot C_R \cdot C_{DF} \cdot C_T}{FS_m}$$

where $C_R$ is the redundancy coefficient of the structural components, here equal to 0.9; $C_{DF}$ is the coefficient that takes into account the service class and the duration of the load application equal to 0.55, 0.65, or 0.85 in cases of permanent, transient, and instantaneous loads; $C_T$ is the coefficient related to the service temperature, here equal to 1; and $FS_m$ is the safety coefficient of the material, which varies depending on the stress.

Similarly, the elasticity modules can be obtained starting from the characteristic values $E_{i,k}$ as:

$$E_{i,d} = E_{i,k} \cdot C_{DE} \cdot C_T$$

Table 4 shows the value of the strengths and the Young moduli used in the structural analysis. For all elements, the geometry has been considered cylindrical with a thickness 1/10 of the diameter.

## 5. Structural Design

A structural calculation is carried out according to the current European and Italian legislation. In particular, the standards UNI EN12810-1:2004 and UNI EN12810-2:2004 [21,22] together with ISO22156:2021 [18] and NTC2018 [23] are used.

### 5.1. Load Analysis

For the load analysis, EN1281-1:2004 is divided into service and out-of-service.

For service loads, the legislation establishes various classes with different values of the load per unit of surface to be applied based on the nature of the work carried out on the scaffolding. In these case studies, since the scaffolding is intended for maintenance and renovation works, service class 2 was adopted. Therefore, the value of the vertical load was considered equal to $q_1 = 1.5$ kN/m$^2$. Furthermore, an additional service load equal to 50% of $q_1$ must be applied on the work surface above and below the considered level. In the absence of wind action, the scaffolding must also be able to support a horizontal load representing operation during use, acting at all the levels where the working area is loaded. A horizontal load not less than 2.5% of the total of the uniformly distributed load $q_1$, or 0.3 kN, whichever is greater, is considered for each bay. Furthermore, the load must be applied at the level of the working surface in both parallel and perpendicular directions to the bay. The loads due to the wind must then be taken into consideration and have been calculated following the current national legislation.

For loads out of service, it is necessary to take into account loads due to snow and wind. European legislation leaves the evaluation of these loads to the various national regulations. In the Italian case, the technical standard NTC2018 is used. The resulting force of the wind action in the out-of-service condition is calculated considering the kinetic pressure of the wind for a return period of 50 years (for the municipality of Bologna, this is 0.391 kN/m$^2$).

For both the in-service and out-of-service conditions, the permanent loads were added, and the load combinations were then conducted. All the structural elements belonging to the scaffolding were designed following NTC2018 at the ultimate limit state (ULS) and at the service limit state (SLS) to verify the safety and functionality for the entire period of the use, which goes from assembly to dismantling. The combinations considered for the analysis are reported in Table 5.

**Table 5.** Different load combinations.

| Denomination | Load Combinations |
| --- | --- |
| 1 | Service condition with the action of the main service loads |
| 2 | Service condition with the action of the main service loads and the secondary wind |
| 3 | Out-of-service condition with the main wind action |
| 4 | Out-of-service condition with primary action snow and secondary wind action |
| 5 | Out-of-service condition with main action wind and secondary snow |

### 5.2. Structural Analysis

The four scaffolding structures (Prototypes 1 and 2 with two or three bay lengths) were pre-dimensioned and then verified. The pre-dimensioning was conducted considering the transoms subject to bending, the standards, and the ledgers at normal stress with the heaviest load of the different load combinations. Subsequently, a model was built in Strauss, and all the verifications recommended by ISO22156:2021 were carried out. The analyses with all the loading conditions were conducted, and the verifications were carried on at the ultimate limit state. Table 6 reports the maximum values of axial forces, shear, and bending moment for the different elements of the structure and the more onerous load condition.

**Table 6.** Maximum values of axial, shear forces, and bending moment for the different elements and the related loading conditions.

| Prototype | Structural Elements | Axial Force [kN] | | Shear Force [kN] | | Bending Moment [kNmm] | |
|---|---|---|---|---|---|---|---|
| Prototype 1 | Standards | 5.30 | ULS 4 | 0.31 | ULS 3/5 | 98.0 | ULS 1 |
| | Ledgers | 0.66 | ULS 5 | 1.40 | ULS 1 | - | - |
| | Transoms | 1.49 | ULS 3 | 2.30 | ULS 1 | 0.70 | ULS 1 |
| | Diagonals | 1.40 | ULS 1 | - | - | - | - |
| Prototype 2 | Standards | 3.92 | ULS 4 | 0.33 | ULS 3/5 | 56.1 | ULS 1 |
| | Ledgers | 0.56 | ULS 5 | 0.94 | ULS 1 | - | - |
| | Transoms | 1.58 | ULS 5 | 1.60 | ULS 1 | 0.37 | ULS 1 |
| | Diagonals | 0.88 | ULS 1 | - | - | - | - |

If a specific verification was not satisfied, the dimensions of the cross section of the culm were enlarged. Where possible, a single bamboo culm was used for a single structural element (with the only exception of the transoms, which are composed of two culms). Tensile, compression, bending, and shear tests were carried out. For standards, the bending test is the most demanding. For the transoms, the most demanding test, which increased in section, was the shear test near the support. Deformability checks were then carried out for the individual bamboo components subject to bending. Specifically, it is checked that the deflections caused by the various actions distributed on the scaffolding are under certain limits imposed by the national regulation NTC-2018. Only the deformations relating to a non-prolonged duration of the loads are evaluated. For elements of wooden origin, the limit relating to the characteristic load combination is equal to 1/300 of the span of the element (§4.4.7 NTC-2018).

In conclusion, Table 7 shows the dimensions and thicknesses of the culms of the various structural elements.

**Table 7.** Diameter and thickness of the structural elements of the scaffolding.

| Prototype | Structural Elements | $D_{min}$ [mm] | $\delta_{min}$ [mm] |
|---|---|---|---|
| Prototype 1 | Standards | 75 | $\cong 8$ |
| | Ledgers | 75 | $\cong 8$ |
| | Transoms | 85 | $\cong 9$ |
| | Diagonals | 50 | $\cong 5$ |
| Prototype 2 | Standards | 65 | $\cong 7$ |
| | Ledgers | 65 | $\cong 7$ |
| | Transoms | 70 | $\cong 7$ |
| | Diagonals | 50 | $\cong 5$ |

*5.3. Verification of the Connection*

The maximum shear force that the connection between standards and ledgers has to take is 1.40 kN, while the connection between standards and transoms is 2.30 kN (divided into two culms).

Considering the preliminary test performed in Section 3, the studied connection could be able to support the calculated shear, but the connections need a more extended experimental campaign.

**6. Resulting Scaffolding Structure**

A list of the required bamboo culms was created. The actual length of the culms was calculated by adding 20 cm to ensure that the connectors could be inserted correctly and safely and considering a maximum length of 4.5 m for transportation without special permission. Figure 10 shows the front and side views of the scaffolding using the three-bay-length model of Prototype 1 as an example. As can be seen from the diagrams,

the connections between two consecutive parallel elements were applied in a 'staggered manner' to avoid weak sections within the structure. Table 8 shows the number and dimensions of all the elements necessary to create the skeleton of the structure.

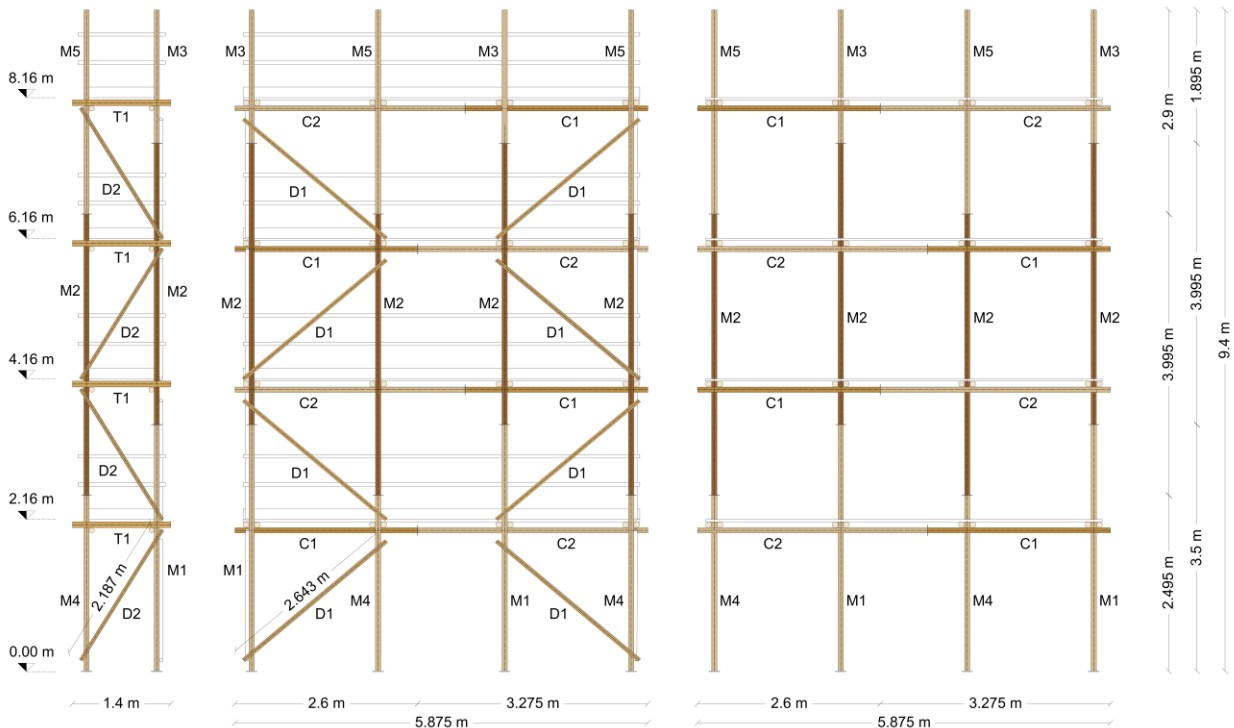

**Figure 10.** Front and side views of the scaffolding using the three-span model of Prototype 1.

**Table 8.** Structural element list of Prototype 1.

| Elements | | Number | Length [m] | $D_{min}$ [mm] | $\delta_{min}$ [mm] |
|---|---|---|---|---|---|
| Standards | M1 | 4 | 3.50 | 75 | $\cong 8$ |
| | M2 | 8 | 4.00 | 75 | $\cong 8$ |
| | M3 | 4 | 1.90 | 75 | $\cong 8$ |
| | M4 | 4 | 2.50 | 75 | $\cong 8$ |
| | M5 | 4 | 2.90 | 75 | $\cong 8$ |
| Ledgers | C1 | 8 | 2.60 | 75 | $\cong 8$ |
| | C2 | 8 | 3.30 | 75 | $\cong 8$ |
| Transoms | T1 | 32 | 1.40 | 85 | $\cong 9$ |
| Diagonals | D1 | 19 | 2.70 | 50 | $\cong 5$ |
| | D2 | 8 | 2.20 | 50 | $\cong 5$ |

## 7. Conclusions

The structure of scaffolding can take on multiple forms based on the dimensions of the building and the various limits and obstacles that may arise in the workplace. Two prototypes were calculated within the European regulatory context, with small dimensions but widely applicable in European urban and extra-urban areas. The structural analysis showed that a single culm could be used for standards, ledgers, and diagonals. For the first prototype, the minimum diameters of the culms varied between 7.5 and 8.5 cm, while for the second prototype, they varied between 6.5 and 7.5 cm. These dimensions may fall within the production range of Italian or European bamboo and are comparable with those

currently used in Hong Kong: 7.5 cm for the Mao Jue type and 4 cm for the Kao Jue type. The advantage of Asian models is that it is possible to use the smaller elements (Kao Jue) thanks to the denser rod system.

The species analyzed in the study is the *Phyllosctachys edulis* (Moso), which can grow in Europe. However, other species that are equally performing and dimensionally comparable could be employed.

A crucial point for bamboo scaffolding is the connection system. This paper proposes a solution that is easy to use, effective, and with a possible quantification of the performance, by employing BAMBUTIX connectors, which are already available on the market.

This paper concludes that bamboo scaffolding could be a sustainable viable option in European countries.

**Author Contributions:** Conceptualization, L.M.; methodology, L.M. and D.A.; software, D.A.; data curation, D.A.; writing—original draft preparation, L.M.; writing—review and editing, L.M. and D.A.; supervision, L.M. All authors have read and agreed to the published version of the manuscript.

**Funding:** This research received no external funding.

**Institutional Review Board Statement:** Not applicable.

**Informed Consent Statement:** Not applicable.

**Data Availability Statement:** The data presented in this study are available on request from the corresponding author.

**Acknowledgments:** CONBAM is acknowledged for providing the connection tested in this paper. Giuseppe Di Lorenzo is acknowledged for his contribution to the tests on the connections during his thesis.

**Conflicts of Interest:** The authors declare no conflicts of interest.

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
