# Peer review of "Bamboo Scaffolding as a European Promising Opportunity: A Structural Feasibility Study"

_sustainability, doi:10.3390/su16020915_

Round 1

Reviewer 1 Report

Comments and Suggestions for Authors

Abstract

I highly recommend to expand abstract and add some numerical results

Intro

I suggest adding small paragraph about bamboo fiber, its chemical composition, & mechanical properties. preferebly in tabulated format to compare it with other natural fibers.

Add small paragraph about lower carbon footprint (LCA) in case bamboo got used.

Can you add few photos for current bamboo scaffolding in Asia?

Comments on the Quality of English Language

Good structured language

Author Response

The authors would like to thank the Editor and the Reviewers for their valuable revision, which strongly contributed to improving the quality of the manuscript. The observations were extremely useful and constructive. They were considered to review the manuscript.

As required by Reviewer 4, the title was slightly changed to better clarify the focus of the research.

The revised parts were marked in the manuscript in red colour.  

Response to Reviewer 1

I highly recommend to expand abstract and add some numerical results

The abstract has been modified and extended

Intro

I suggest adding small paragraph about bamboo fiber, its chemical composition, & mechanical properties. preferebly in tabulated format to compare it with other natural fibers.

The introduction has been extended considering the different characteristics of bamboo without a specific section reminding the reader of the huge literature on the topic.

Add small paragraph about lower carbon footprint (LCA) in case bamboo got used.

We thank the reviewer for the comment. We agree that this is a crucial aspect that should be detailed. A small paragraph has been added in the Introduction.

Can you add few photos for current bamboo scaffolding in Asia?

A photo of the connection of a scaffolding structure from Hong Kong has been added

Reviewer 2 Report

Comments and Suggestions for Authors

Is an interesting proposal, with bamboo culms to elaborate scaffolding reinforced by using steel connections. Structural analysis for bamboo culms and steel unions is well described and discussed. That paper structure is consistent between approach, development, analysis and conclusions.

Use of metallic unions is different to scaffolding conventional in Asia, that use fiber laces. Proposal uf use bamboo species that can grow in Europe is a good idea.

I consider convenient evaluate the corrosion resistance to metllic unions to prolongued times. Also, may be tested polimeric materials to unions, due to their less weigth and major durability.  

References are enough, but may be improved including recents documents.

Tables 1, 2, 5, 7 and 8 can be adjusted the width of columns to improve visual presentation.
Figure 2, is anoted A), b) and c), but text only do mention to a) and b).

In line 113, says "steal", must be "steel".

Author Response

The authors would like to thank the Editor and the Reviewers for their valuable revision, which strongly contributed to improving the quality of the manuscript. The observations were extremely useful and constructive. They were considered to review the manuscript.

As required by Reviewer 4, the title was slightly changed to better clarify the focus of the research.

The revised parts were marked in the manuscript in red colour.   

Response to Reviewer 2

Is an interesting proposal, with bamboo culms to elaborate scaffolding reinforced by using steel connections. Structural analysis for bamboo culms and steel unions is well described and discussed. That paper structure is consistent between approach, development, analysis and conclusions.

Use of metallic unions is different to scaffolding conventional in Asia, that use fiber laces. Proposal uf use bamboo species that can grow in Europe is a good idea.

I consider convenient evaluate the corrosion resistance to metllic unions to prolongued times. Also, may be tested polimeric materials to unions, due to their less weigth and major durability.  

References are enough, but may be improved including recents documents.

Some references have been added

Tables 1, 2, 5, 7 and 8 can be adjusted the width of columns to improve visual presentation.
Figure 2, is anoted A), b) and c), but text only do mention to a) and b).

We thank the Reviewer for the suggestions, the Tables and the Figures are changed accordingly.

In line 113, says "steal", must be "steel".

Thank you very much for highlighting this typo, we corrected it

Reviewer 3 Report

Comments and Suggestions for Authors

A brief summary: The paper performed tests on bamboo scaffolding that utilizes steel connections. The bamboo investigated is one that can grow in Europe. The tests were conducted to demonstrate the feasibility of different scaffolding bay lengths and heights and different steel connection types. The paper proposed that bamboo scaffolding with steel connections could be a sustainable viable option in European countries. The main contributions are the results from the tests performed.  

General concept comments
Article: There are no areas of weakness. No method/tool was developed in the study, it was just different tests conducted to prove that the bamboo employed, and its steel connections were adequate for use in a scaffold.
Review: The manuscript only needs minor spelling revisions for full completeness. The knowledge gap that was identified and the solution that was proposed are of relevance, and will be of more relevance if indeed the overall cost for bamboo scaffolding is lesser than that of conventional scaffolding without compromising structural capacity and durability. All references identified by the authors are appropriate for this study.  

Specific comments

Aside spelling errors, everything else looks good.

Author Response

The authors would like to thank the Editor and the Reviewers for their valuable revision, which strongly contributed to improving the quality of the manuscript. The observations were extremely useful and constructive. They were considered to review the manuscript.

As required by Reviewer 4, the title was slightly changed to better clarify the focus of the research.

The revised parts were marked in the manuscript in red colour.  

Response to Reviewer 3

A brief summary: The paper performed tests on bamboo scaffolding that utilizes steel connections. The bamboo investigated is one that can grow in Europe. The tests were conducted to demonstrate the feasibility of different scaffolding bay lengths and heights and different steel connection types. The paper proposed that bamboo scaffolding with steel connections could be a sustainable viable option in European countries. The main contributions are the results from the tests performed.  

General concept comments 
Article: There are no areas of weakness. No method/tool was developed in the study, it was just different tests conducted to prove that the bamboo employed, and its steel connections were adequate for use in a scaffold. 
Review: The manuscript only needs minor spelling revisions for full completeness. The knowledge gap that was identified and the solution that was proposed are of relevance, and will be of more relevance if indeed the overall cost for bamboo scaffolding is lesser than that of conventional scaffolding without compromising structural capacity and durability. All references identified by the authors are appropriate for this study.  

Specific comments

Aside spelling errors, everything else looks good.

Thank you very much for your kind comment, the text has been revised

Reviewer 4 Report

Comments and Suggestions for Authors

This work tries to demonstrate how bamboo scaffolding can be a viable option in Europe, and gives a very preliminary description. The authors should concern the following issues,

l  The necessity for conducting this research should be clearly presented. There is no logic in the Introduction part.

l  The title of this paper is too large, the reader cannot know what they want to do in this work.

l  What are the limitation and uncertainty in the tests? What is the measurement error?

l  Only limited conditions are considered. Are they representative?

l  The analysis on the data is very poor. Moreover, there is a lack of comparative analysis with the previous research.

l  The content of this paper should be enriched, it is at the level of a conference paper for the current version.

Comments on the Quality of English Language

Minor editing of English language required

Author Response

The authors would like to thank the Editor and the Reviewers for their valuable revision, which strongly contributed to improving the quality of the manuscript. The observations were extremely useful and constructive. They were considered to review the manuscript.

As required by Reviewer 4, the title was slightly changed to better clarify the focus of the research.

The revised parts were marked in the manuscript in red colour.  

Response to Reviewer 4

This work tries to demonstrate how bamboo scaffolding can be a viable option in Europe, and gives a very preliminary description. The authors should concern the following issues,

l  The necessity for conducting this research should be clearly presented. There is no logic in the Introduction part.

We thank the Reviewer for the comment the introduction has been largely reviewed.

l  The title of this paper is too large, the reader cannot know what they want to do in this work.

The title has been changed to better focalize the topic of the paper on the structural issues.

l  What are the limitation and uncertainty in the tests? What is the measurement error?

l  Only limited conditions are considered. Are they representative?

We agree with the Reviewer but the tests are preliminary, as stated in the text, the idea is to justify the use of that kind of connection in the scaffolding structure. A study on this kind of connection can be further detailed.

l  The analysis on the data is very poor. Moreover, there is a lack of comparative analysis with the previous research.

To the knowledge of the authors, there are no other tests on this kind of connection.

l  The content of this paper should be enriched, it is at the level of a conference paper for the current version.

We enriched the text to improve the paper.

Round 2

Reviewer 4 Report

Comments and Suggestions for Authors

The authors have improved the manuscirpt significantly.